# Research on Panoramic Stitching Algorithm of Lateral Cranial Sequence Images in Dental Multifunctional Cone Beam Computed Tomography

**DOI:** 10.3390/s21062200

**Published:** 2021-03-21

**Authors:** Junyuan Liu, Xi Li, Siwan Shen, Xiaoming Jiang, Wang Chen, Zhangyong Li

**Affiliations:** 1Medical Electronics and Information Technology Engineering Research Center, Chongqing University of Posts and Telecommunications, Chong Qing 400065, China; s180501011@stu.cqupt.edu.cn (J.L.); s180501004@stu.cqupt.edu.cn (S.S.); jiangxm@cqupt.edu.cn (X.J.); s180131193@stu.cqupt.edu.cn (W.C.); 2Foundation Department, Chongqing Medical and Pharmaceutical College, Chongqing 401331, China; lixixyz@126.com; 3School of Communication and Information Engineering, Chongqing University of Posts and Telecommunications, Chongqing 400065, China

**Keywords:** lateral cephalogram, Gaussian mixture model, image registration, image fusion

## Abstract

In the design of dental multifunctional Cone Beam Computed Tomography, the linear scanning strategy not only saves equipment cost, but also avoids the demand for patients to be repositioned when acquiring lateral cranial sequence images. In order to obtain panoramic images, we propose a local normalized cross-correlation stitching algorithm based on Gaussian Mixture Model. Firstly, the Block-Matching and 3D filtering algorithm is used to remove quantum and impulse noises according to the characteristics of X-ray images; Then, the segmentation of the irrelevant region and the extraction of the region of interest are performed by Gaussian Mixture Model; The locally normalized cross-relation is used to complete the registration with the multi-resolution strategy based on wavelet transform and Particle Swarm Optimization algorithm; Finally, image fusion is achieved by the weighted smoothing fusion algorithm. The experimental results show that the panoramic image obtained by this method has significant performance in both subjective vision and objective quality evaluation and can be applied to preoperative diagnosis of clinical dental deformity and postoperative effect evaluation.

## 1. Introduction

Oral disease is common and frequently occurring in all ages. With the improvement of medical the level, oral imaging has been widely used in preoperative diagnosis and postoperative evaluation of orthodontics, dental implant [1]. Oral imaging can be divided into three-dimensional Cone Beam Computed Tomography (CBCT) imaging and two-dimensional Digital Radiography (DR) imaging. The two-dimensional lateral cephalogram has important clinical value in the diagnosis of oral deformity, the reliability of the upper respiratory tract, the hyoid bone and the soft palate, and the diagnosis of head and cervical spine [2,3]. It can not only help doctors better understand the structure and relative position of the patient’s cranial, maxillary, face and teeth, but also be widely used in the evaluation of the effect of orthodontic treatment [4]. Therefore, the addition of lateral cranial imaging in the design of oral CBCT instrument can expand the function and improve the cost performance.

In order to reduce the cost of the instrument and avoid the demand for patients to be repositioned when switching imaging types in CBCT imaging, the planar detector with a small field of view is used to obtain sequential images through linear scanning. Then, the image stitching algorithm is used to continuously stitch the sequence images to get the panoramic image. Image stitching methods have made great achievements in many fields, but some gaps still exist in continuous stitching of small size. How to stitch a large number of continuous small dimensions without obvious features into a panorama is a difficult but hot topic in current research. A large number of image registration-based stitching methods have been proposed, which are basically divided into five categories [5].

Phase correlation method: Fast Fourier transform (FFT) is applied to the image, and then the translation vector between the two images can be directly calculated by their mutual power spectrum, so that the image registration can be realized. Yan song et al. [6] improved the phase correlation method by combining it with threshold segmentation. Yunyun Dong et al. [7] assumed the noise is a mixed Gaussian distribution, performed rank one matrix decomposition on the phase correlation matrix to complete the image registration. Xiaohua Tong et al. [8] integrated the advantages of Hoge’s method and the RANSAC algorithm and avoided the corresponding shortfalls of the original phase correlation method.

Feature-based method: The key to the success of this method is feature point extraction and matching. Tian Zhang et al. [9] proposed an improved SURF operator by calculating the normalized gray-level difference and second-order gradient in the neighborhood. Herng-Hua Chang et al. [10] improved the Scale Invariant Feature Transform (SIFT) operator, feature slope calculation, feature point grouping, the and outlier removal and transformation were adopted. SK sharma et al. [11] utilized AKAZE to detect feature points, obtained corresponding matching pairs by using K-NN algorithm, and removed the false matched points by MSAC.

Intensity information-based method: This method directly uses the intensity information of two images and determines the best transformation parameters by measuring the similarity. S Song et al. [12] proposed a peripheral mutual information (PMI) maximization method for image registration, which makes use of the closed-form solution for the Shannon entropy. M Pan et al. [13] adopted Renyi’s quadratic mutual information as the similarity criterion between the reference and floating images, and the Simplex method, as multi-parameter optimization one, is used to search the optimal registering values.

Hybrid-based method: In order to synthesize the advantages of different methods, many hybrid-based registration methods have been proposed. M song et al. [14] combined the method based on SIFT feature and mutual information to achieve remote sensing image registration from Coarse-to-fine. Ruitao Feng et al. [15] developed a robust algorithm by combining and localizing feature- and area-based methods. Han et al. [16] aimed to accurately register Optical and TIR images by Using SURF and Local Phase Correlation.

Deep-learning-based method: Deep learning has achieved good performance in feature extraction and parameter registration output. Dosovitskiy et al. [17] trained the convolutional neural network with unlabeled data, and the performance of the extracted feature descriptor was better than that of the SIFT descriptor. Z Yang et al. [18] used a pre-trained VGG network layer to generate a feature descriptor while preserving the convolution information and local features. Cao X et al. [19] used the deformation site as a label and applied supervised learning to output registration parameters. G Wu et al. [20] adopted unsupervised learning and directly input the registration pair into the network to obtain the deformation field. The method based on deep learning requires a large number of manual annotation samples to optimize the learning process [21]. As far as we know, there is no large public database to annotate lateral cephalic sequence images. Therefore, the traditional method is chosen to explore a suitable mathematical method for continuous stitching of small-size images.

Most of the existing image stitching methods are based on feature extraction. For example, L Deng et al. [22] proposed a stitching method based on global alignment model, Kim D T et al. [23] proposed a combination of ROI and reduction technology for endoscopic panoramic video stitching, and Qu Z et al. [24] proposed a stitching method based on binary tree and estimated overlap region. However, the essence of these methods is based on the successful extraction of feature points of the image to be stitched. For the oral cranial sequence images, it is difficult to extract enough feature points from the images to be stitched, and the stability of continuous stitching cannot be guaranteed.

In view of these difficulties, a local normalized cross-correlation stitching algorithm based on Gaussian Mixture Model (GMM) is suggested. The primary contributions of our study are summarized as follows:

BM3D algorithm is applied to remove the quantum and impulse noise of the X-ray sequence image before image stitching.

In order to eliminate the influence of the irrelevant regions in X-ray image stitching, a local normalized cross-correlation registration based on GMM is proposed.

The source image is decomposed into a three-layer image pyramid by wavelet transform [25], and the multi-resolution strategy is combined with the particle swarm optimization algorithm to optimize registration parameters and improve the stitching accuracy.

To process the overlapping area of adjacent images, we combined the distance from the pixel to the boundary of the overlapping area with a weighted average [26], which can make the overlapping area transition naturally.

In the Section 2, we will introduce the principle of lateral cranial imaging with multi-functional CBCT device and explain the properties of images and why image stitching is used. The design of stitching algorithm based on local normalized cross-correlation is presented in Section 3. In Section 4, we will first introduce our experimental dataset, and we will present the results of experiments. Section 5 concludes our work, which can provide a clinical basis for oral diagnosis.

## 2. Multi-Functional Oral CBCT Devices

The traditional multifunctional oral CBCT device is shown in Figure 1a. It consists of a Lateral cranial imaging module (red box 1) and CBCT imaging module (red box 2). When taking a 3D image of the oral cavity, CBCT module is used to carry out the ring scan and then reconstruct this image on a computer. The patient needs to be moved to the lateral cranial module for repositioning during lateral cranial radiographs. This imaging strategy results in the cost of the equipment being expensive, as at least two sets of X-tube and flat plate detectors are required.

The data in this paper came from the new multi-functional CBCT, as shown in Figure 1b. 3D oral images and lateral cranial sequence images can be obtained at the same time by positioning the patient once. The lateral cranial imaging module is shown in Figure 1c, and we only need to rotate the detector of CBCT imaging by 90 degrees and adopt the strategy of combining small field detector with linear scanning to obtain the serial images of lateral cranial position. The X-ray tube sends X-rays to transit the head to a small field flat panel detector, which converts the X-ray image directly into an analog signal, which A/D converts into A digital signal. Due to the size limitations of the X-ray ball tube and sensor, only a small vertical slice of the skull was recorded at each imaging session. Therefore, in order to obtain the lateral cephalogram of the entire head model, the X-ray ball tube and detector were moved from point a to point b, and multiple images were taken consecutively. This design strategy not only avoids the need to reposition the patient during the acquisition of lateral cephalogram, but also saves equipment costs by requiring only one set of X-ball tubes and flat plate detector.

A partial image sequence obtained by linear scanning is shown in Figure 1c, and each image is 2232(height) × 60(width) pixels in size, and all images are partially overlapped with adjacent ones. The data in this paper came from the human head model, and a total of 480 small-size images were obtained by linear scanning. Sequential images need to be processed by a stitching algorithm to obtain panoramic images before they can be used for clinical diagnosis.

## 3. Design of Stitching Algorithm

In this study, the feature-based approach is not applicable to the data in this paper. The main difficulty is that it is difficult to extract effective features from the 60-pixel-wide images, especially in the soft tissue areas without teeth and bones. SIFT feature extraction operator with strong robustness and rotation invariance [27] was selected to extract and match features of some sequence images. As shown in Figure 2, The number of feature points extracted is not enough to calculate the transformation matrix between adjacent images, and some feature points are mismatched.

In this paper, intensity information is directly used as the direct feature, which has the characteristics of simplicity, high precision, and strong anti-jamming ability. The flow chart of the overall algorithm is shown in Figure 3, which is mainly composed of four parts: Image denoising, image segmentation, image registration, and image fusion.

### 3.1. Image Denoising

Image noise refers to the unnecessary or redundant interference information existing in image data, and the noise of X-ray image mainly comes from system structure, charge diffusion, photon fluctuation, analog-to-digital conversion, and so on [28]. According to the category of noise, it mainly includes quantum noise and impulse noise.

Quantum noise is the random spatial fluctuation of X-ray quantum according to Poisson distribution law, and its magnitude is related to the radiation dose of the ray source and the absorption performance of the plate detector. In general, it is considered that the quantum fluctuation noise of the radiography system obeys the Poisson distribution, which is expressed as follows:(1)$$p(z)=\frac{\lambda^{z}}{z!}e^{-\lambda }\;\lambda >0$$

Impulse noise is the randomly distributed noise generated on the image by the excitation action of X-ray scattering on the flat panel detector. Impulse noise can be described by the following probability density function:(2)$$p(z)=\{\begin{array}{cc} P_{a} & z=a\\ P_{b} & z=b\\ 0 & others \end{array}$$

The image sequence needs to be de-noised before the image is stitched, mean filtering, median filtering, Gaussian filtering, and NL-Means algorithm are widely used denoising methods, and BM3D algorithm has excellent performance for denoising X-ray images among many denoising algorithms [29]. The BM3D algorithm was proposed by Dabov K et al. [30]. The flow chart of the BM3D algorithm is shown in Figure 4. It is generally divided into two steps; each step includes similar block grouping, collaborative filtering, and aggregation. The algorithm makes a basic estimate before making a final estimate of the image.

Similar block grouping: For each target block, look for a maximum of MAXN1 similar blocks in the vicinity, this process can be expressed as follows:(3)$$G(P)=\left\{Q\;:\;d(P,\;Q)\leq \;\tau^{step}\right\}$$
where d(P,Q) is the Euclidean distance between the two blocks, after sorting from smallest to largest, take the preceding maxN1 to form a three-dimensional array.

Collaborative filtering: The two-dimensional transformation of each two-dimensional block in the three-dimensional matrix is carried out, and then the one-dimensional transformation is carried out in the third dimension of the matrix. After transformation, the parameter less than the super parameter is set to 0 by means of the hard threshold. At the same time, the number of non-zero components counted as a reference for subsequent weights. This process is as follows:(4)$$Q(P)=T^{-1}{}_{3Dhard}(\gamma (T_{3Dhard}(Q(P))))$$

Aggregation: After the inverse transformation of these graph blocks, the algorithm puts them back to the original position, uses the number of non-zero components to calculate the stack weight, and finally divides the stacked graph by the weight of each point to get the basic estimated image.

### 3.2. Image Segmentation

X-rays are penetrative, but the density and thickness of human tissue vary, and as X-rays penetrate bone (including teeth) and muscle, the denser the tissue, the better it absorbs the X-ray, so the imaging process produces different images. Four images were randomly selected from the sequence to observe their gray histogram distribution, as shown in Figure 5.

The histogram reflects the probability of the occurrence of a certain grayscale value in the image. The grayscale value of the image in this paper is distributed between 0 and 14,000, and most of the grayscale histograms of sequence images are of peak structure. After analysis, we know that the area with smaller gray value in the histogram is bone imaging, while the part with larger gray value (red box) belongs to air and soft tissue imaging. For the registration of sequence images, bone imaging is regarded as the target region of interest, but when there is no bone imaging, soft tissue is regarded as the region of interest. An appropriate image segmentation method is used to extract the target region for subsequent registration steps.

In order to avoid the influence of the background region, the image is segmented by the clustering method. The pixel value of the image is taken as the clustering element, so that all the points in the image are divided into two categories to achieve the segmentation of the target area and the background area. Gaussian Mixed model (GMM) is used to estimate the probability density distribution of the sample, and each Gaussian model represents a class. GMM (Gaussian mixture model), by the weighted combination of several Gaussian distribution models, can be used to fit any type of distribution, and this process can be expressed as follows:(5)$$GMM(i)=\sum_{j=1}^{j=c}\varphi_{j}\frac{1}{\sqrt{2\pi \sigma }}\times \exp\left[-\frac{{\left(h(i)-u_{j}\right)}^{2}}{2\sigma_{j}{}^{2}}\right]$$
where C represents the number of Gaussian distributions, h(i) represents the gray value of the pixel point, $\varphi_{j}$ represents the weight of the j Gaussian distribution, $u_{j}$ and $\sigma_{j}^{2}$ represent the mean and variance of the Gaussian distribution.

The maximum expectation (EM) algorithm is used to estimate the parameters of each Gaussian function, and the segmentation of foreground and background regions is completed. It is a kind of optimization algorithm for maximum likelihood estimation through iteration, which consists of alternating Expectation-step and Maximization-step. Expectation-step calculates the expectation of the implicit variable based on the initial value of the parameter or the model parameters of the last iteration, which is expressed as:(6)$$Q_{i}(z^{\left(i\right)}):=p(z^{\left(i\right)}|\;x^{\left(i\right)};\theta )$$

Maximization-step maximizes the likelihood function to obtain the new parameter value, which is expressed as:(7)$$\theta :=\arg\;\underset{\theta }{\max}\sum_{i}\sum_{z^{\left(i\right)}}Q_{i}(z^{\left(i\right)})\log\frac{p(x^{\left(i\right)},z^{\left(i\right)};\theta )}{Q_{i}(z^{\left(i\right)})}$$

Algorithm 1 summarizes the use of Gaussian mixture model combined with EM algorithm to segment sequence images.
**Algorithm 1** Gaussian mixture model combined with EM algorithm for image segmentation**Input:** Original sequence images. **Step1:** Initializes the parameters of the GMM $u_{j}$, $\sigma_{j}$, $\pi_{j}$. **do**
**{**
**Step2:** Calculate the value of the posterior probability per Equation (6). **Step3:** Maximize the likelihood function to get the new $u_{j}$, $\sigma_{j}$, $\pi_{j}$ per Equation (7). } **while** (The logarithm likelihood function does not converge) **Step4:** Pixels are allocated using the maximum posterior probability criterion. **Output:** Segmentation results.

Four images were selected from the sequence of images for segmentation, and the segmentation results are shown in Figure 6.

Figure 6a,b images without bone areas are shown, and the segmentation method can accurately extract soft tissue areas. In Figure 6c,d, most of the bone areas can be completely segmented, and the boundary of the extracted target region was clear and continuous.

### 3.3. Image Registration

Adjacent image registration is the key to panoramic image generation technology. Its essence is to find out the position of the corresponding point in the reference image through a certain matching strategy, and then determine the transformation relationship between two images. The image registration process in this paper is shown in Figure 7, which consists of the wavelet transform, spatial transformation, interpolator, similarity measure, and optimization algorithm.

#### 3.3.1. Wavelet Transform

Image pyramid is a representation of the multi-scale image. An image pyramid is a set of images with different resolutions arranged in a pyramid shape. The bottom of the pyramid is a high-resolution representation of the image, while the top is a low-resolution approximation. In image registration, the multi-resolution strategy can not only shorten the registration time, but also reduce the influence of image noise on the registration results [31]. Wavelet transform has the characteristics of multi-resolution and has higher frequency resolution in the low-frequency part. The Wavelet transform of the two-dimensional image and the wavelet function is expressed as:(8)$$W_{s}f(a,b)=\int_{R}\int_{R}f(x,y)\frac{1}{s^{2}}\times \psi (\frac{a-x}{s},\frac{b-y}{s})d_{x}d_{y}$$

Two-layer wavelet decomposition is performed on the original image, as shown in Figure 8, to obtain the low-frequency part A1 and other B1 of the image. The low-frequency part A1 can retain the features of the source image, and then wavelet decomposition is performed on A1 to form an image pyramid composed of A2, A1, and the original image, as shown in Figure 9.

#### 3.3.2. Spatial Transformation

To successfully register two adjacent oral cephalic images $X_{1}\left(a_{1},b_{1}\right)$ and $X_{2}\left(a_{{}_{2}},b_{2}\right)$, the key is to determine the mapping relationship between them: $P:\left(a_{1},b_{1}\right)\to \left(a_{2},b_{2}\right)$, make each point in $X_{1}$ and $X_{2}$ correspond one to one, that is, a corresponding point in two images represents the same position. The image transformation mode can be divided into local, global, and displacement field. Global transformation means that the transformation of the whole image can be represented by the same parameters. Different regions in local transformation can have different parameters. Displacement field transformation shows that each pixel in the image is independent of parameter conversion.

The global mapping model is adopted in this paper, which is more suitable for automatic registration, which can be defined as follows:(9)T(x)=A(x−c)+t+c
where A represents matrix, C represents the Center of the image, and T represents translation. Since the data are taken from the human head model, object deformation is not involved in the imaging process, that is, the distance and parallel relation of the corresponding points of adjacent images remain unchanged, so the rigid body transformation is selected as the global transformation model [32]. $\left(x_{1},y_{1}\right)$ is the coordinate before the transformation, and $\left(x_{2},y_{2}\right)$ is the new coordinate after the transformation, and the conversion format can be described as: (10)$$\left[\begin{array}{l} x_{2}\\ y_{2} \end{array}\right]=s\left[\begin{array}{l} \cos\theta -\sin\theta \\ \sin\theta \;\cos\theta \end{array}\right]\;\left[\begin{array}{l} x_{1}\\ y_{1} \end{array}\right]+\left[\begin{array}{l} d_{x}\\ d_{y} \end{array}\right]$$
where S is the scaling factor, θ is the rotation angle, and the translation in the x and y directions are respectively $\left[d_{x}\;d_{y}\right]$.

#### 3.3.3. Interpolator

In general, the digital image displays the image through the gray value, and the discrete pixel coordinates are usually integers, while the pixel coordinates of the image to be registered are generally not integers after the spatial transformation. The image interpolation is used to solve the problem that the pixel point of the image is not integer after the spatial transformation, and the image pixel value at integer position is obtained by interpolation. In this paper, a bilinear interpolation algorithm is selected to process the image. The core idea is shown in Figure 10.

First, we interpolate the X and Y directions respectively to get two points, R1 and R2, and then we interpolate the points R1 and R2 to get the value of P, which is expressed as:(11)f(x,y)=f(0,0)(1−x)(1−y)+f(1,0)x(1−y)+f(0,1)(1−x)y+f(1,1)xy
where f(x,y) is the value of the interpolated point P, f(0,0), f(0,1), f(1,0) and f(1,1) are the values of points Q1, Q2, Q3, and Q4, respectively.

#### 3.3.4. Similarity Measure

The similarity measure is a quantitative measure to indicate the degree of similarity between the reference image and the image to be registered. The selection of similarity measure function directly determines the reliability and validity of image registration. Image registration is to find the optimal parameters to ensure that the similarity or difference between images to be registered can reach the maximum value, so selecting the appropriate similarity measurement function is helpful to improve the accuracy of registration. The most commonly used similarity measures are mutual information, normalized cross-correlation, and so on. Among them, the normalized cross-correlation has the characteristics of high accuracy and good stability in registration, but it is easily affected by the irrelevant area in the image during registration.

In this paper, a similarity measurement function based on GMM and Normalized Cross-Correlation (NCC) for ROI is proposed, which avoids the influence of the irrelevant background region in registration. GMM-NCC is defined as:(12)$$GMM-NCC=-\frac{\sum_{i,j}{\langle I_{ref_{ROI}}\left(i,j\right)-{\bar{I}}_{ref_{ROI}},I_{reg_{ROI}}\left(i,j\right)-{\bar{I}}_{reg_{ROI}}\rangle }^{2}\;}{\sum_{i,j}\;{\left|I_{ref_{ROI}}\left(i,j\right)-{\bar{I}}_{ref_{ROI}}\right|}^{2}\;{\left|I_{reg_{ROI}}\left(i,j\right)-{\bar{I}}_{reg_{ROI}}\right|}^{2}\;}$$
where $I_{ref_{ROI}}$ and $I_{reg_{ROI}}$ respectively represent the regions of interest of the fixed images and images to be registered, ${\bar{I}}_{ref_{ROI}}$ and ${\bar{I}}_{reg_{ROI}}$ represent their mean values, i and j are the coordinates of pixels. When GMM−NCC=1, it means that the two registered images are completely unrelated; When GMM−NCC=−1, it means that the registration effect between two images is the best.

Figure 11a is the measure curve of the classical normalized cross-correlation measure function, and Figure 11b is the normalized cross-correlation measure curve based on the GMM proposed. The results show that the measurement function proposed in this paper has higher accuracy for the same sequence image registration.

#### 3.3.5. Optimizer

Using the optimization algorithm to optimize the registration parameters can speed up the search process of parameters and reduce the registration time of sequence images. At present, the Powell algorithm, conjugate gradient algorithm, simulated annealing algorithm, and genetic algorithm are widely used [33,34]. The first two algorithms are local optimization algorithms, which depend on the selection of initial values. Although the latter two algorithms are global optimization algorithms, the computational process is complex, and the convergence speed is slow. In summary, the particle swarm optimization algorithm is adopted in this paper, which is an optimization method based on the swarm intelligence method and is very suitable for solving the global optimal solution. Suppose that m particles form a population in an-dimensional space, and the position of the ith particle is defined as:(13)$$x_{i}=\left(x_{i1},x_{i2},\ldots ,x_{in}\right)\;(i=1,2,\ldots ,m)$$

The current position is substituted into the similarity measure to measure the advantages and disadvantages of the particle position. If it is not optimal, the update of the particle position is defined as:(14)x(n+1)=x(n)+v(n+1)
where v(n+1) is the flight speed of the particle at the next update, which is defined as:(15)$$v(n+1)=wv(n)+c_{1}r_{1}(pbest-x(n))+c_{2}r_{2}(gbest-x(n))$$
where v(n) is the current particle’s flight speed, w is the inertial weight, $c_{1}$ and $c_{2}$ are the learning factors, $r_{1}$ and $r_{2}$ are the random numbers in the range of (0,1).

Algorithm 2 summarizes the registration process of adjacent sequence images.
**Algorithm 2** Image registration algorithm in this paper**Input:** Target regions of fixed and to be registered images: R, F. **Step1:** Two three-layer image pyramids are formed by wavelet transform of two images: $R_{i},F_{i}$, (i=1,2,3). **Step2:** registration from low resolution to high resolution registration. i=3.    for each $R_{i}$ and $F_{i}$
i∈1,2,3 do      **Step3:**
$F_{i}$ performs spatial transformation per Equation (9).      **Step4:**
$F_{i}$ is interpolated after the spatial transformation per Equation (11).      **Step5:**
$R_{i}$ and $F_{i}$ measure GMM-NCC per Equation (12).        if the value of GMM-NCC is optimal do          if *i* = 1 do output final parameters.          else do i=i−1.            Output the current registration parameters and return to step 3.        else do Optimize using the PSO algorithm and then g return to Step 3. **Output:** The displacement parameters $\left[d_{x}\;d_{y}\right]$ and rotation parameters θ.

### 3.4. Image Fusion

Image fusion refers to the process of merging the registration results into a natural and uniform image according to a specific fusion method under the same scene. It can retain all the information of the registration image to the maximum extent and is the last key step in the image stitching process. In the application of image Mosaic, the function of image fusion is to make the stitching image smooth and natural transition at the boundary of the overlapping area without damaging the original image quality, and to eliminate the stitching gap and ghost problems caused by different contrast and exposure.

Firstly, the fixed image and the image to be registered are transformed into a unified coordinate system, and then the image is transformed according to the displacement parameters obtained from the image registration. In order to save the computation time and make the overlapping area more natural, the weighted fusion algorithm is used to deal with the stitching problem, which is expressed as:(16)Pixel=kPixel_L+(1−k)Pixel_R
where k is the weighted coefficient, which is defined as $k=\frac{d_{1}}{d_{1}+d_{2}}$. $d_{1}$ and $d_{2}$ are the distances from each point to the boundary of the overlapping region, respectively.

## 4. Result and Discussion

The experiments were operated using python3.7 on a computer with i5-8000CPU and 8 GB RAM. Firstly, the original image sequence is denoised by the BM3D algorithm, and then the target region of the sequence image is extracted by GMM to remove the influence of the irrelevant background region in the registration. Furthermore, the GMM-NCC measure function proposed in this paper is optimized by combining the multi-resolution strategy based on wavelet variation with the particle swarm optimization algorithm, and the optimal registration parameters are output. Finally, the overlapping area is processed by image fusion.

The image data we used are from the imaging of a human head model. The sequence of images consists of 480 images, each of which has a size of 2320 (height) × 60 (width) pixels and 65,536 grey levels.

Before image registration, we used BM3D algorithm to remove quantum noise and impulse noise of all original sequence images. As shown in Figure 12, the BM3D algorithm can well remove the noise in the original image and avoid the impact on the subsequent steps.

The essence of image registration is the problem of parameter optimization. In order to verify the accuracy of the registration algorithm in this paper, a group of adjacent images in the image sequence are selected to measure their real displacement parameters using image software, and the algorithm in this paper is compared with three different algorithms for registration. In order to avoid the randomness of single registration, it is assumed that k∈[1,50], and the registration parameter $x_{k}$ in the *x* direction is selected, and the registration error is defined as $error=x-x_{k}$. k is taken as the abscissa and k as the ordinate to draw a registration error statistical graph, as shown in Figure 13.

Figure 13 shows the statistical chart of registration error comparison between the proposed method and the three different algorithms. As shown in Figure 11a, Yuan Huang et al. [35] used the Normalized Cross-Correlation algorithm to register adjacent images, the fluctuation range of registration error is [8, −6]. As shown in Figure 11b, Xiaoping Liu et al. [36] combined local self-similarity and mutual information (LSS-MI) to register multi-sensor images and the fluctuation range of registration error is [6, −6]. As shown in Figure 11b, Yang H et al. [37] proposed a registration method combining SIFT feature-based and phase correlation, and the fluctuation range of registration error is [4, −3]. The proposed algorithm is more stable in the registration process. Compared with the other three methods, the fluctuation range is [1, −1], and the registration accuracy reaches the sub-pixel level, which is closer to the real displacement parameters.

In order to directly observe the influence of registration error on image Mosaic, Figure 14a,b is selected to adopt the above registration method for image stitching. The same position of the two images is marked with red dots. Figure 14c,d and Figure 14e are respectively the panoramic images obtained by the above three registration algorithms, and it can be seen that the marking points do not overlap. Figure 14f is the panoramic image of the registration method in this paper, and the overlap of the marking points indicates that the estimation of registration parameters is accurate.

As shown in Figure 15a, the continuous stitching strategy was adopted in this paper to complete the stitching of 480 original sequence images. Figure 15b is an image of the anterior part of the skull obtained from serial image stitching, which is mainly used for the diagnosis and postoperative evaluation of oral deformity. Figure 15c is a stitching image of the posterior portion of the skull, which is used to diagnose the spine. Figure 15b,c obtained the final cranial lateral panoramic image of the oral cavity through a round of stitching.

In order to verify the excellent ability of the algorithm in this paper to stitching images of lateral cranial sequences, we compared it with three different stitching methods. The stitching results of the algorithm in this paper are shown in Figure 16a; panoramic images have no stitching gaps and ghosts and the information remains intact during the stitching process. Ref. [38] uses SIFT to extract features for stitching, resulting in information loss in boneless areas of lateral cranial images, as shown in Figure 16b. Ref. [39] uses the combination of normalized cross-correlation and threshold method to stitching the image, but there was much false stitching and ghosting in the complex texture area, as shown in Figure 16c. Ref. [14] uses the strategy of SIFT combined with mutual information (SIFT-MI) to make up for the shortcoming of failing to extract feature points, however there are obvious ghosts and stitching gaps in some areas, as shown in Figure 16d. Ref. [40] uses an improved phase correlation (PC) algorithm and weighted blending to generate panoramic images, and there are a lot of information missing and stitching errors in the image, as shown in Figure 16e. Ref. [41] uses a new and improved algorithm based on the Accelerated KAZE (A-KAZE) Features, and there are significant stitching errors in the cervical stitching of the image, as shown in Figure 16f.

The performance of the proposed algorithm can be measured more objectively by using image quality evaluation parameters. The entropy value can determine the information amount in the image. For the same data source, the larger the entropy value of the Mosaic image is, the more complete the retained information will be, and the better the performance of the Mosaic algorithm will be, which can be expressed as: (17)$$IE\left(x\right)=-\sum_{i=1}^{n}P\left(a_{i}\right)*\log P\left(a_{i}\right)$$

The standard deviation reflects the degree of dispersion between the pixel value of the image and the mean value. The higher the standard deviation, the better the image quality, which can be expressed as:(18)$$SD=\sqrt{\frac{1}{M\times N}\sum_{i=1}^{M}\sum_{j=1}^{N}{\left(P(i,j)-u\right)}^{2}}$$

The average gradient represents the average value of gray scale transformation, reflecting the clarity of the stitching image. The larger the value is, the more obvious the detail of the image is, and the clearer the image is, which is expressed as:(19)$$AG=\frac{1}{\left(M_{r}-1)(N_{c}-1\right)}\sum_{i=1}^{M_{r}-1}\sum_{j=1}^{N_{c}-1}\sqrt{\frac{1}{2}[{\left(h(i,j)-h(i+1,j)\right)}^{2}+{\left(h(i,j)-h(i,j+1)\right)}^{2}]}$$

For continuous stitching of sequence images, the success rate is also an important index to measure the algorithm, which represents the stability of the algorithm in practical application. Five algorithms were used to conduct 100 stitching experiments on the sequential images, and the stitching success rate of lateral cephalic panorama could be obtained statistically.

Through the above discussion, it is obvious that the stitching algorithm proposed in this paper can complete the continuous stitching of dental lateral cephalic sequence images. The image quality evaluation results obtained are shown in Table 1. Compared with the other Five algorithms, although the running time of the algorithm in this paper is slightly longer than that of some algorithms, the image quality after stitching is better. The maximum information entropy of the method in this paper indicates that the retained information is the most complete. The maximum standard deviation and evaluation gray value indicate that the brightness transformation of the image obtained by the algorithm in this paper is uniform and the image details remain intact. The success rate of the proposed algorithm in this article is higher than that of other algorithms, which indicates that the proposed algorithm has good stability.

In X-ray imaging, the higher the density of the tissue, the more obvious the absorption of X-rays. In order to make the images more detailed for clinical diagnosis, therefore we need to reverse color manipulation of the stitching panoramic image, as shown in Figure 17.

## 5. Conclusions

In the field of image stitching, image registration and image fusion have a great influence on the stitching result. Image stitching methods are mainly based on feature extraction or intensity information. The deep learning method has achieved a good result in feature extraction, and due to the limitations of the part of the imaging area and lack of annotated dataset of lateral cephalogram, continuous stitching of the sequence images is difficult. The methods of NCC and NMI algorithms based on intensity information have low stitching accuracy for small size X-ray images with high dynamic range, and it is easy to appear stitching gaps and ghosts.

This paper makes improvements from the perspective of image registration; the BM3D algorithm is used to remove the impact of quantum noise and impulse noise of the original X-ray image and innovatively proposes the normalized cross-correlation registration method of ROI based on GMM segmentation, while the irrelevant background region is removed successfully. The multi-resolution strategy based on wavelet decomposition realizes accurate registration from low resolution to high resolution, and PSO is used to optimize the displacement parameters. After the image registration, the adjacent images are fused by combining the distance between the pixel points and the boundary of the overlapping region. Panoramic images of the lateral dental skull were obtained after successive rounds of stitching. Compared with other methods, the stitching panoramic image performed better in both subjective visual and objective evaluation., the image information remains intact during the stitching process, the image details are clear, and there are no stitching gaps and ghosts. The experimental results show that the proposed algorithm can complete continuous stitching of sequential images, which is of great significance for lateral cephalic imaging based on multifunctional dental CBCT.

## Figures and Tables

**Figure 1 sensors-21-02200-f001:**
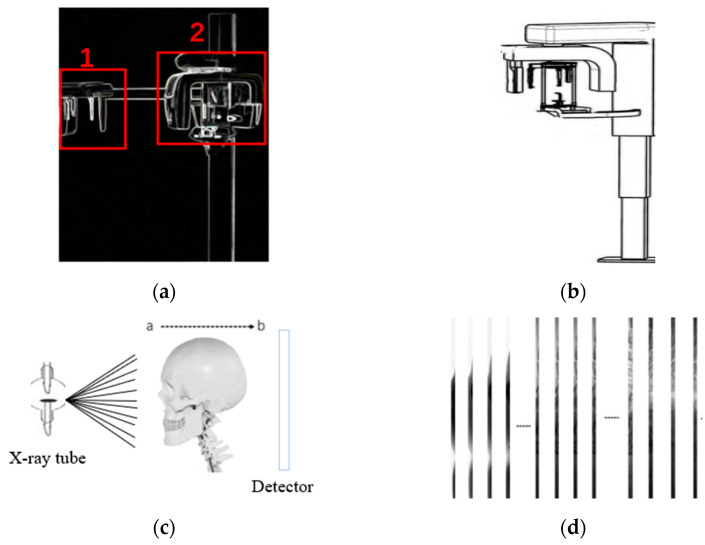
Multifunctional Cone Beam Computed Tomography (CBCT) dental cranial lateral imaging. (**a**) The traditional multi-functional oral CBCT device; (**b**) new multi-functional CBCT equipment; (**c**) imaging process; (**d**) partial sequence image.

**Figure 2 sensors-21-02200-f002:**
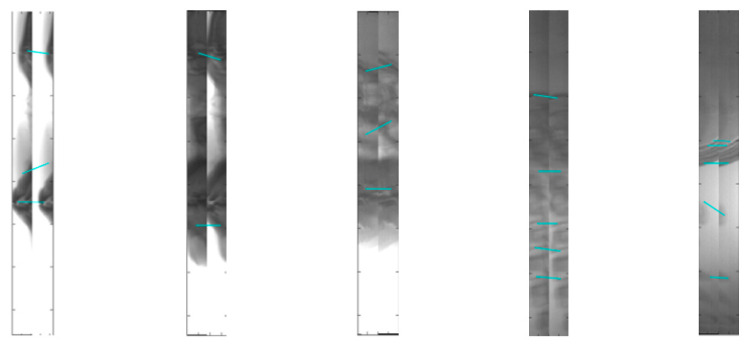
SIFT feature extraction and matching results.

**Figure 3 sensors-21-02200-f003:**
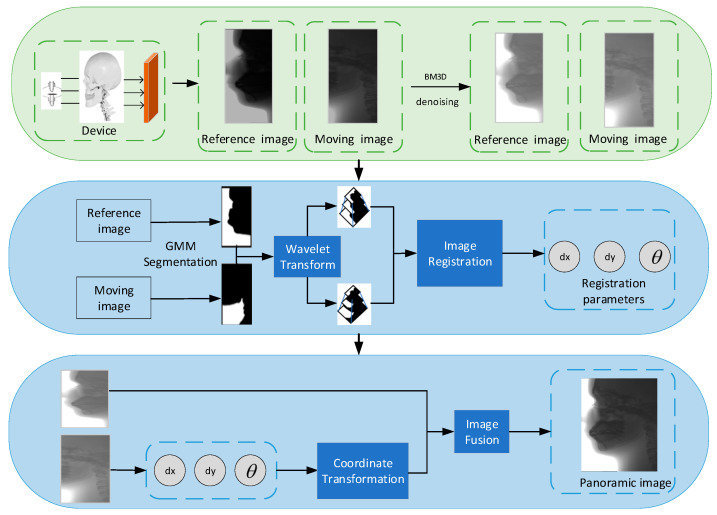
Flowchart of the proposed method for lateral cranial image stitching.

**Figure 4 sensors-21-02200-f004:**
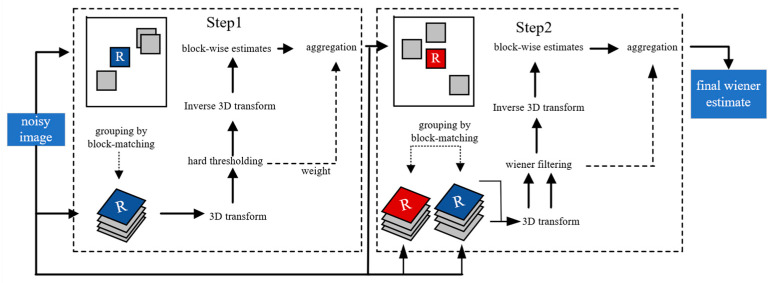
BM3D algorithm flow chart.

**Figure 5 sensors-21-02200-f005:**
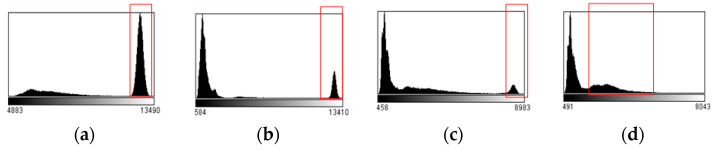
Histogram of part of the image sequence. (**a**) Sequence image 1; (**b**) sequence image 2; (**c**) sequence image 3; (**d**) sequence image 4.

**Figure 6 sensors-21-02200-f006:**
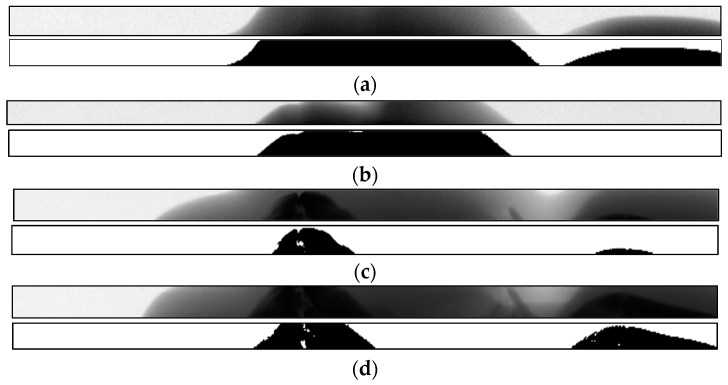
Segmentation results of four selected images in the sequence. (**a**) Sequence image 1; (**b**) sequence image 2; (**c**) sequence image 3; (**d**) sequence image 4.

**Figure 7 sensors-21-02200-f007:**
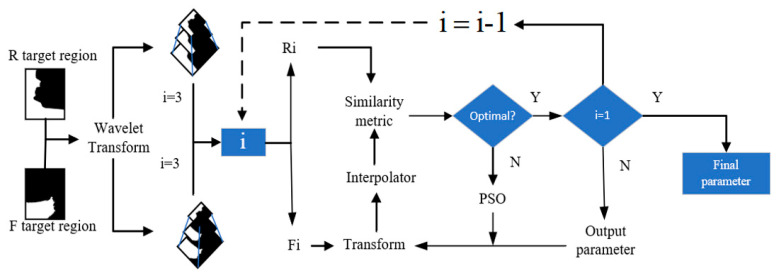
Image registration flow chart.

**Figure 8 sensors-21-02200-f008:**
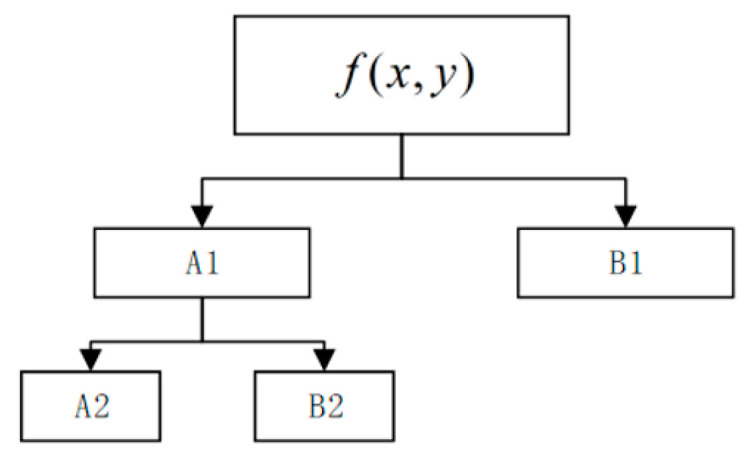
Two-layer wavelet decomposition tree.

**Figure 9 sensors-21-02200-f009:**
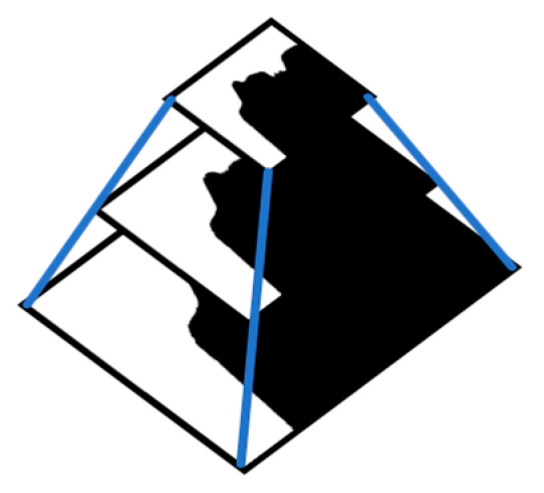
Three layers of image pyramid.

**Figure 10 sensors-21-02200-f010:**
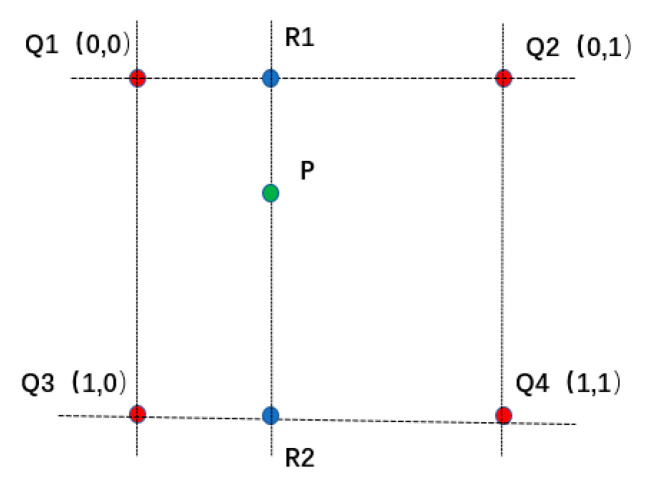
Bilinear interpolation schematic diagram.

**Figure 11 sensors-21-02200-f011:**
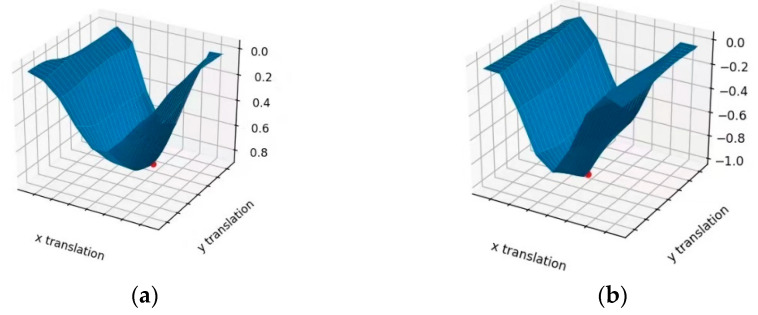
A graph of the similarity measurement function. (**a**) Normalized cross correlation; (**b**) local normalized cross-correlation based on Gaussian mixture model.

**Figure 12 sensors-21-02200-f012:**
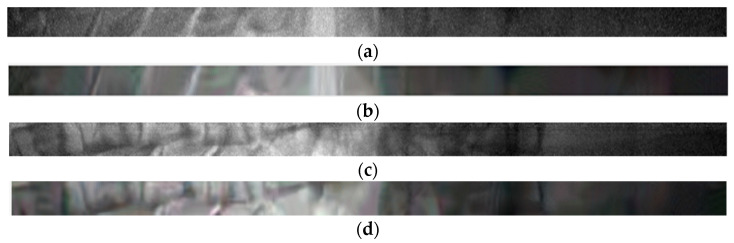
The denoising effect of BM3D on sequential image. (**a**) Original image 1, (**b**) The denoising result of image 1, (**c**) Original image 2, (**d**) The denoising result of image 2.

**Figure 13 sensors-21-02200-f013:**
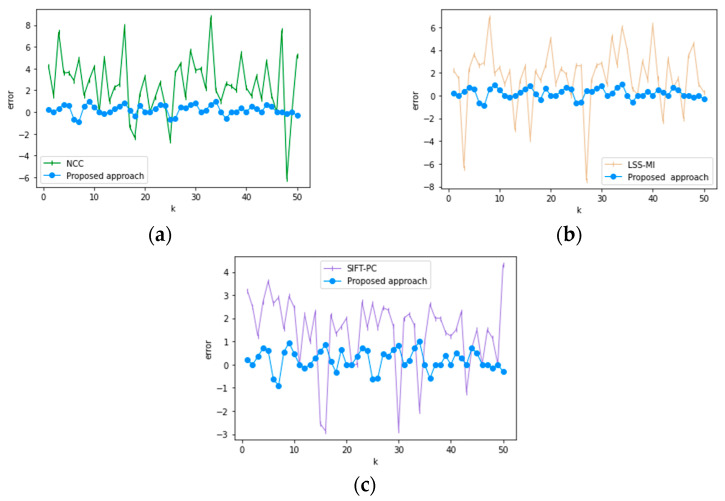
Statistical graph of registration error. (**a**) NCC and the proposed approach, (**b**) LSS-MI Comparison with the algorithm in this paper, (**c**) SIFT-PC and the algorithm in this paper.

**Figure 14 sensors-21-02200-f014:**
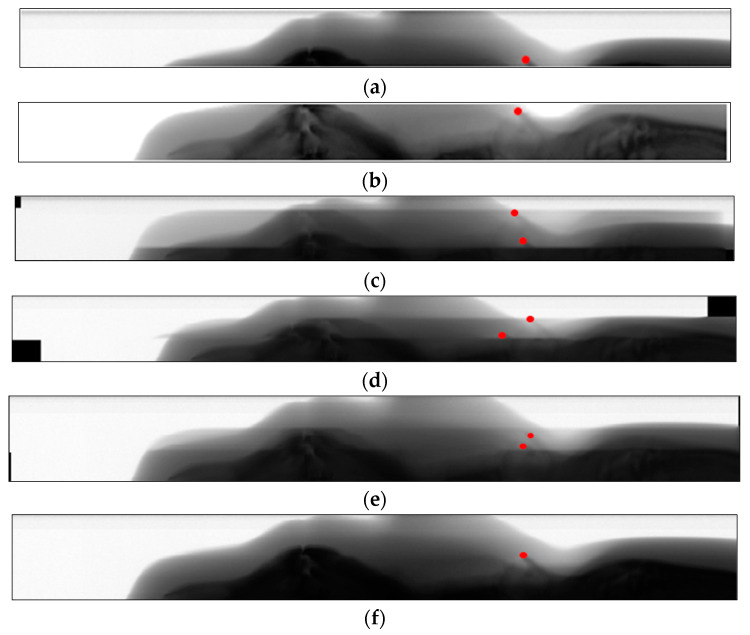
Error observation of registration. (**a**) Image 1 to be stitched, (**b**) Image 2 to be stitched, (**c**) NCC, (**d**) LSS-MI, (**e**) SIFT-PC, (**f**) this article.

**Figure 15 sensors-21-02200-f015:**
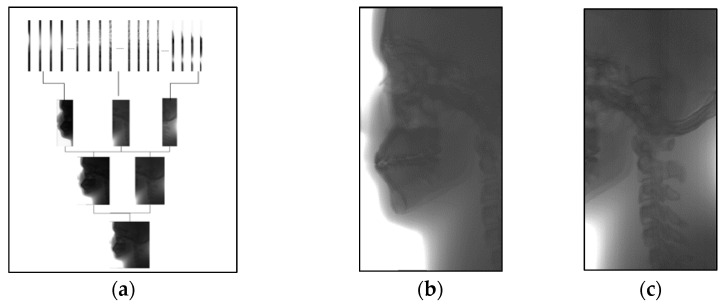
Stitching process of lateral cephalogram. (**a**) The image sequence is stitched in multiple rounds, (**b**) results of anterior skull stitching, (**c**) results of posterior skull stitching.

**Figure 16 sensors-21-02200-f016:**
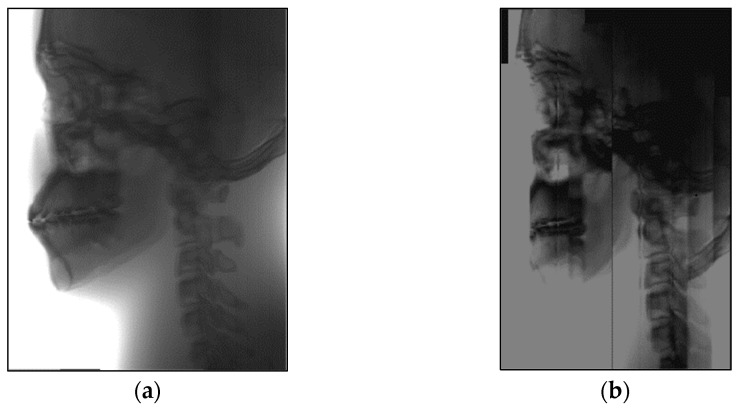
Sequence image stitching results by different algorithms. (**a**) This article, (**b**) SIFT-PSO, (**c**) NCC, (**d**) SIFT-NMI, (**e**) PC, (**f**) A-KAZE.

**Figure 17 sensors-21-02200-f017:**
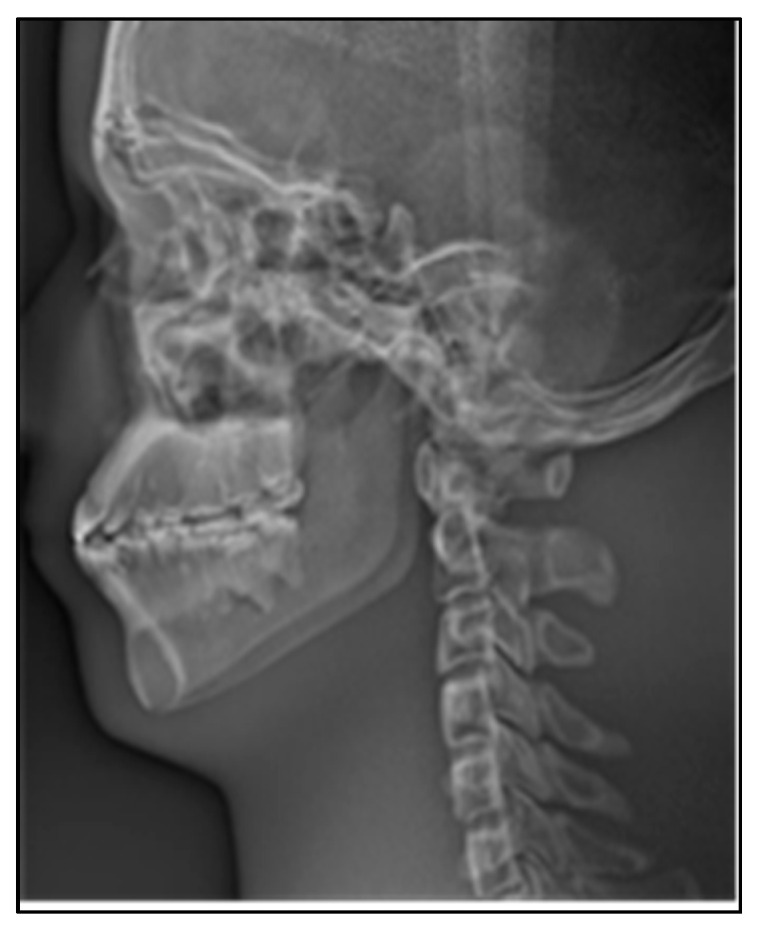
Lateral cephalogram after reverse color manipulation.

**Table 1 sensors-21-02200-t001:** Different stitching algorithms quality evaluation parameters.

Methods	IE	SD	AG	Time	Success Rate
SIFT-PSI	5.0631	42.0415	1.4327	78.6542 s	68%
NCC	6.0254	73.1553	1.4493	106.0612 s	73%
SIFT-NMI	6.0554	68.0053	1.557	83.5381 s	78%
PC	4.1567	40.2985	1.3645	97.9457 s	76%
A-KAZE	5.9683	67.8642	1.4765	62.2938 s	59%
This article	6.1711	74.6245	1.6913	98.3258 s	98%

## Data Availability

The data presented in this study are available on request from the corresponding author. The data are not publicly available because it is for the benefit of the equipment manufacturer.

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
