# Peer review of "Research on Panoramic Stitching Algorithm of Lateral Cranial Sequence Images in Dental Multifunctional Cone Beam Computed Tomography"

_sensors, 2021, doi:10.3390/s21062200_

Round 1

Author Response

Dear reviewer:

I am very grateful to your comments for the manuscript. According with your advice, we amended the relevant part in manuscript. Some of your questions were answered below.

C001-Many grammatical or typographical errors have been revised.

C002- L75-L80 A fifth image registration type is added.

C003- The organization of this manuscript is added to the end of the introduction.

C004-Figure3- A flow chart of the proposed method is presented.

C005- The original table 1 was deleted.

C006-L513-L515 Two groups of comparative experiments were added.

C007-Corrected errors in references.

Thank you and all the reviewers for the kind advice.

Sincerely yours.

Reviewer 2 Report

The manuscript Sensors-1135865, submitted as an original research paper in the field of computer vision, exploits a stitching algorithm in radiological dental imaging. Specifically, the authors opt to use a cone-beam dental CT (CBCT) device to create a lateral cephalogram for orthodontic purposes by a 90° twisting of the CBCT detector. The cone-beam detector, however, is orthogonal wider-than-tall and does not cover the anteroposterior head dimension. The authors suggest as a solution the acquisition of a lateral cone beam scan with slab-stitching to create a lateral head-neck panorama and provide a stitching algorithm.     

The topic is not novel. Still, cost-effective improvements in dental imaging are a rising research field with merit for publication.   

Although the manuscript suggests a straight-forward plausible algorithm to image stitching, concerned with issues of computational speed with a wavelet-based data compaction strategy, there are several issues that should be tackled before publication.  

Major comments:

C001 A professional language editing is required, both in the field of use of English, text structure and clarity. The manuscript will profit from the consultation of a native speaker with experience in the field. Please provide a receipt of the proof editing for the interest of the editor.

C002 The authors work in stitching algorithms to create a panoramic cephalogram, acquired by the CBCT device by twisting the detector panel by 90°. To understand this aim, one has to read the whole paper till Figure 14 – then it becomes clear what the authors mean.

The text needs an extended reformulation of the aims and purposes.

Moreover, elaborating about the CBCT (title, abstract, L122-130 a.s.o.) disorientates even an experienced reader. Please focus the formulation of methods, aims, and scopes to the relevant information.  

C003 Please compare the radiation dosis difference of your method (panoramic cephaliogram) with the dosis of a digital lateral cephalogram (gold standard), and elaborate on this in the discussion.

C004 – L227 and Figure 5: “most histograms are bimodal”. In Figure 5 the authors present two bimodal and two at least trimodal to quadrimodal histograms. Please provide a complete histogram analysis of an image sequence using another metric to support your statement

C005 – L264 and L260, Figure 6. In line 260, “three images are selected”, in L264 “6 images are selected” and Figure 3 shows 3 SEQUENCES (not images). Please decide.

C006 – L434m L436 and L438 : “Figure 11” correct to Figure 12

C007 – Compare your methods to the results of previous studies, especially to the ones that implement artificial intelligence.

C008 – Add to Figure 12 (comparison to other methods) a graphical example like Fig 14 to accompany the line plots

C009 – Table 2: explain IE, SD AG. If SD is the standard deviation and IE the image entropy, no statistical significance is assumed between all compared methods. Please comment on it.  

C010 – It would be useful for the reader to compare the computational time of your algorithm to other solutions

C011 – The citation list could be enriched by highly-ranked recent literature on image stitching, including protocols that should be compared with the current suggestion, as well as at least one inspired review highlighting the challenges in the field on image stitching, extending beyond the field of dental imaging.

Suggested literature (not compulsory)  

 [1]J. Chalfoun et al., “MIST: Accurate and Scalable Microscopy Image Stitching Tool with Stage Modeling and Error Minimization,” Sci Rep, vol. 7, no. 1, p. 4988, Jul. 2017, doi: 10.1038/s41598-017-04567-y.

[2]C. Chen, R. Kojcev, D. Haschtmann, T. Fekete, L. Nolte, and G. Zheng, “Ruler Based Automatic C-Arm Image Stitching Without Overlapping Constraint,” J Digit Imaging, vol. 28, no. 4, pp. 474–480, Aug. 2015, doi: 10.1007/s10278-014-9763-3.

[3]L. Deng, X. Yuan, C. Deng, J. Chen, and Y. Cai, “Image Stitching Based on Nonrigid Warping for Urban Scene,” Sensors (Basel), vol. 20, no. 24, Dec. 2020, doi: 10.3390/s20247050.

[4]L. Kang, Y. Wei, J. Jiang, and Y. Xie, “Robust Cylindrical Panorama Stitching for Low-Texture Scenes Based on Image Alignment Using Deep Learning and Iterative Optimization,” Sensors (Basel), vol. 19, no. 23, Dec. 2019, doi: 10.3390/s19235310.

[5]D. T. Kim, V. T. Nguyen, C.-H. Cheng, D.-G. Liu, K. C. J. Liu, and K. C. J. Huang, “Speed Improvement in Image Stitching for Panoramic Dynamic Images during Minimally Invasive Surgery,” J Healthc Eng, vol. 2018, p. 3654210, 2018, doi: 10.1155/2018/3654210.

[6]A. Li, X. Liu, W. Gong, W. Sun, and J. Sun, “Prelocation image stitching method based on flexible and precise boresight adjustment using Risley prisms,” J Opt Soc Am A Opt Image Sci Vis, vol. 36, no. 2, pp. 305–311, Feb. 2019, doi: 10.1364/JOSAA.36.000305.

[7]A. Miettinen, I. V. Oikonomidis, A. Bonnin, and M. Stampanoni, “NRStitcher: non-rigid stitching of terapixel-scale volumetric images,” Bioinformatics, vol. 35, no. 24, pp. 5290–5297, Dec. 2019, doi: 10.1093/bioinformatics/btz423.

[8]B. A. Millis and M. J. Tyska, “High-Resolution Image Stitching as a Tool to Assess Tissue-Level Protein Distribution and Localization,” Methods Mol Biol, vol. 1606, pp. 281–296, 2017, doi: 10.1007/978-1-4939-6990-6_18.

[9]C. Murtin, C. Frindel, D. Rousseau, and K. Ito, “Image processing for precise three-dimensional registration and stitching of thick high-resolution laser-scanning microscopy image stacks,” Comput Biol Med, vol. 92, pp. 22–41, Jan. 2018, doi: 10.1016/j.compbiomed.2017.10.027.

[10]Z. Qu, J. Li, K.-H. Bao, and Z.-C. Si, “An Unordered Image Stitching Method Based on Binary Tree and Estimated Overlapping Area,” IEEE Trans Image Process, May 2020, doi: 10.1109/TIP.2020.2993134.

[11]Z. Qu, X.-M. Wei, and S.-Q. Chen, “An algorithm of image mosaic based on binary tree and eliminating distortion error,” PLoS One, vol. 14, no. 1, p. e0210354, 2019, doi: 10.1371/journal.pone.0210354.

[12]S. Saalfeld, “Computational methods for stitching, alignment, and artifact correction of serial section data,” Methods Cell Biol, vol. 152, pp. 261–276, 2019, doi: 10.1016/bs.mcb.2019.04.007.

[13]O. Svystun, L. Schropp, A. Wenzel, J. M. de C. E. S. Fuglsig, M. H. Pedersen, and R. Spin-Neto, “Prevalence and severity of image-stitching artifacts in charge-coupled device-based cephalograms of orthodontic patients,” Oral Surg Oral Med Oral Pathol Oral Radiol, vol. 129, no. 2, pp. 158–164, Feb. 2020, doi: 10.1016/j.oooo.2019.07.004.

[14]O. Svystun, A. Wenzel, L. Schropp, and R. Spin-Neto, “Image-stitching artefacts and distortion in CCD-based cephalograms and their association with sensor type and head movement: ex vivo study,” Dentomaxillofac Radiol, vol. 49, no. 3, p. 20190315, Mar. 2020, doi: 10.1259/dmfr.20190315.

[15]F. Yang, Y. He, Z. S. Deng, and A. Yan, “Improvement of automated image stitching system for DR X-ray images,” Comput Biol Med, vol. 71, pp. 108–114, Apr. 2016, doi: 10.1016/j.compbiomed.2016.01.026.

[16]J. Zaragoza,  null Tat-Jun Chin, Q.-H. Tran, M. S. Brown, and D. Suter, “As-Projective-As-Possible Image Stitching with Moving DLT,” IEEE Trans Pattern Anal Mach Intell, vol. 36, no. 7, pp. 1285–1298, Jul. 2014, doi: 10.1109/TPAMI.2013.247.

[17]Y. Zhang, Y.-K. Lai, and F.-L. Zhang, “Content-Preserving Image Stitching with Piecewise Rectangular Boundary Constraints,” IEEE Trans Vis Comput Graph, vol. PP, Jan. 2020, doi: 10.1109/TVCG.2020.2965097.

[18]A. Zomet, A. Levin, S. Peleg, and Y. Weiss, “Seamless image stitching by minimizing false edges,” IEEE Trans Image Process, vol. 15, no. 4, pp. 969–977, Apr. 2006, doi: 10.1109/tip.2005.863958.

MINOR COMMENTS

Abstract and title: Explain abbreviations throughout the text. Abbreviations are not recommended in the title

L27: “oral disease”; which oral disease? Oral diseases vary from cancer to allergic swelling. The authors mean dental & gum diseases, maybe? Please specify.

L90 and L92: The normalized cross-correlation algorithm is once described as “insensitive to brightness change” and then “affected by the high pixel brightness”. Kindly clarify.

Author Response

Dear reviewer:

I am very grateful to your comments for the manuscript. According with your advice, we amended the relevant part in manuscript. Some of your questions were answered below.

C001-Many grammatical or typographical errors have been revised.

C002-L124-L144 We focused the method, purpose, and scope more on relevant information and removed some of the irrelevant information.

C003-As for the radiation dose, since the equipment is currently under development, the experimental radiation dose is 3USV, close to the clinical gold standard of 1-5USV. In order to ensure the accuracy of the data, the paper does not give the detailed radiation dose.

C004-L224-L232 A complete histogram analysis method is provided and the segmentation method is discussed

C005-L259-The problem of "three images are selected" is modified, and four images are re-selected for the segmentation experiment.

C006-“Figure 11” correct to Figure 12.

C007-L513-L515 Two groups of comparative experiments were added. However, due to the particularity of the data in this paper, the method based on deep learning is not suitable for continuous stitching of lateral cranial images.

C008-L473-L479 A graphic example is added in the original figure 12 to facilitate intuitive observation of the impact of registration on the stitching results。

C009-L530-L534 the evaluation index of success rate is added in the algorithm evaluation.

C010-Table1-The solution time of the proposed algorithm is compared with other methods.

C011-L94-L101 In the introduction, the literature on image stitching is added, and the shortcomings of image stitching in lateral cranial sequence are analyzed.

C012-Abbreviations have been changed in the title and abstract.

C013-L29-L31 types of oral diseases are specified as dental implants and orthodontic treatment.

C014-Modified the instructions on normalized cross-correlation.

Thank you and all the reviewers for the kind advice.

Sincerely yours.

Round 2

Reviewer 1 Report

All my questions have been solved.

Reviewer 2 Report

Dear authors, 

I was glad to receive the revised version of your manuscript. The reviewer's points are adequately covered in the corrected version. However, the text was not edited by a native speaker, and various grammar mistakes hamper the scientific quality. Please refer to the editor for further advice in this topic.   

Sincerely,